# A Comparison of the Transient Effect of Complex and Core Stability Exercises on Static Balance Ability and Muscle Activation during Static Standing in Healthy Male Adults

**DOI:** 10.3390/healthcare8040375

**Published:** 2020-10-01

**Authors:** Ho-Jin Shin, Jin-Hwa Jung, Sung-Hyeon Kim, Suk-Chan Hahm, Hwi-young Cho

**Affiliations:** 1Department of Health Science, Gachon University Graduate School, Incheon 21936, Korea; sports0911@hanmail.net (H.-J.S.); 315201@hanmail.net (S.-H.K.); 2Department of Occupational Therapy, Semyung University, Jecheon 27136, Korea; otsalt@nate.com; 3Graduate School of Integrative Medicine, CHA University, Seongnam 13488, Korea; 4Department of Physical Therapy, Gachon University, Incheon 21936, Korea

**Keywords:** postural sway, complex exercise (CE), balance, muscle activation, core stability exercise (CSE)

## Abstract

Balance ability is a necessary exercise factor required for the activities of daily living. This study investigated the effects of short-term complex exercise (CE) and core stability exercise (CSE) on balance ability and trunk and lower-extremity muscle activation on healthy male adults. Twenty-nine healthy male adults were included. All performed CE and CSE for 1 min each; the exercise order was randomized. The primary and secondary outcomes were balance ability and muscle activation, respectively. In balance ability, CE showed a significant difference in all variables in both eye-opened and eye-closed conditions compared with the baseline (*p* < 0.05). In comparisons among exercises, the path length and average velocity variables showed a significant decrease in the eye-opened condition, and the path length variable showed a significant decrease in the eye-closed condition (*p* < 0.05). In muscle activation, CE showed a significant increase in the gluteus medius (Gmed) and decrease in the rectus femoris (RF), tibialis anterior (TA), and RF/biceps femoris (BF) ratio in the eye-opened condition compared to the baseline and a significant decrease in RF and RF/BF ratio in the eye-closed condition (*p* < 0.05). Both CE and CSE improved the static balance ability. Furthermore, muscle activation significantly increases in Gmed and decreases in the RF, TA, and RF/BF ratio. Therefore, we recommend including CE in an exercise program that has the purpose of improving static balance ability.

## 1. Introduction

Balance ability is essential in performing activities of daily living (ADLs) such as walking, sitting, and standing as well as sports activities. Balance training is considered necessary for ADLs as it facilitates improved work ability in standing and sitting positions and prevents the elderly from falling [1]. Furthermore, in modern life, because of increased participation in leisure sports that require static standing balance, such as surfing, snowboarding, archery, and shooting, and as delicate work has increased, balance training is also emphasized to improve sports playing ability and work efficiency [2,3]. Static and dynamic balance ability is required to improve in the short and mid–long term according to each situation. Balance training is prescribed to prevent and treat injuries caused by poor balance ability, and part of this is core stability exercise. The plank, bridge, bracing maneuver, modified curl-up, and quadruped bird dog, known collectively as core exercise, are typical examples [4,5,6].

In 2010, Kaji et al. reported that CSE transiently improved balance ability [7]. Furthermore, they suggested that it can be utilized as a warm-up, especially in sports such as archery or shooting that require static standing balance. Plank and bridge exercises are widely known to be effective exercise methods to strengthen trunk stability and improve balance ability [8,9]. However, because these exercises have to be performed while lying or facing down, there is a spatial limitation to performing them in public spaces or before a sports match. Moreover, Kaji et al. report that plank and bridge exercises elicited a significant change in sway in the mediolateral direction during static balance. This result indicates that these exercises do not effectively respond to perturbation in various directions, including the anteroposterior (AP) direction, which can occur in ADLs.

Various exercise methods to improve balance ability such as tiptoe standing, tandem stance, weight shifting, and bracing maneuvers, in addition to traditional CSE, are commonly suggested in the literature [5,10,11,12]. However, unlike CSE, these exercises not only have no immediate effects on balance ability but are also included in parts of a balance training program, so the net effect of each exercise is unclear. The composition of an effective exercise to improve balance is also unclear. Therefore, it is necessary to solve these problems and suggest an exercise method that can overcome the spatial limitation of existing exercise methods. Therefore, the purpose of this study was to compare the transient effect of CSE, which is traditionally used, and complex exercise (CE), which is newly developed and combines the existing exercises, static balance ability and muscle activation.

## 2. Materials and Methods

### 2.1. Participants

This study was conducted for 3 months at Gachon University. Fifty-eight university students responded to a campus announcement, and 29 met the inclusion criteria and participated in the experiment (Figure 1). The inclusion criteria were as follows: (1) healthy male adults aged between 20–29 years, (2) those who have never performed core exercises before, and (3) those able to perform the exercises by following instruction. The exclusion criteria were as follows: (1) those with orthopedic diseases in the lower extremity; (2) those with neurological diseases; (3) those with diseases affecting proprioceptive sense, vision, and the vestibular organ; (4) those with flat or high-arch feet; and (5) those who exercise twice a week for more than 30 min per session [10]. Prior to the start of the study, every participant provided written informed consent. The study was approved by the Gachon University Institutional Review Board (1044396-201612-HR-109-01). The experimental procedure of this study was conducted according to the Declaration of Helsinki.

### 2.2. Study Design

We used a crossover design. All 29 participants performed CE and CSE, and to exclude the order effect, a complete counterbalancing method was used; moreover, exercise order was randomized. Measurements were obtained by a physical therapist with a master’s degree and clinical experience of more than 5 years who was blinded to the study.

### 2.3. Experimental Procedures

To be able to perform the exercise properly and with the correct movement, the participants performed a familiarization session 1 week prior to the experiment. Before participation, we obtained each individual’s general characteristics, including sex, age, height, and weight. First, surface electromyography (EMG) electrodes were attached to the selected muscles of the participants. The surface EMG and force plate were synchronized. The static balance ability and muscle activation were measured in eye-opened and eye-closed conditions while standing on the force plate. Each exercise (CE and CSE) was performed for 1 min, and static balance ability and muscle activation were measured identically as a baseline after exercises.

### 2.4. Training Protocol

#### 2.4.1. Complex Exercise

This exercise consisted of a bracing maneuver, which activates the core muscles, and tiptoe and tandem exercises. The composed exercises were selected as exercises that can improve balance and are easy to perform without space constraints [5,11,12,13,14]. The participants were instructed to perform tandem and tiptoe exercises, both with a bracing maneuver.

##### Bracing Maneuver

It was performed by instructing them to “focus on the abdominal muscles and hold their breath after exhalation” and then maintain the abdominal isometric contraction [15]. For the proper performance of the bracing maneuver, a researcher confirmed core muscle contraction and trunk lateral expansion through palpation between the 12th rib and iliac crest in the axillary line before starting the exercise. To maintain bracing maneuver during tandem exercise and tiptoe exercise, the therapist confirmed the EMG activity and provided verbal cues to the participant.

##### Tandem Exercise

The participant had a tandem stance posture with the gaze frontward, arms at the pelvis side, and a minimum distance of 5 cm between the front and rear foot. Weight shifting was performed in an anteroposterior direction like a pendulum movement while taking the tandem stance. Weight shifting was performed at a constant rhythm using a smartphone metronome application (Tempo Lite, Frozen Ape Pte. Ltd., Singapore) adjusted to participants’ preferred speed (1–2 Hz). During weight shifting, the weight shifting range was moved until only the first and second metatarsal heads of both feet touched the ground when moving forward, and only the calcaneus of both feet touched when moving backward. The trunk maintained the bracing maneuver without flexion or extension movement (Figure 2). This exercise was performed for 30 s, and the participant continued weight shifting in an anteroposterior direction.

##### Tiptoe Exercise

The participant stood upright with feet shoulder-width apart, staring straight ahead with arms positioned at the sides. Then, the participant was only instructed to raise his heel. During the exercise, the therapist instructed that the bracing maneuver be maintained (Figure 3). This exercise was performed for 30 s, and the participant kept his heel out of contact with the ground.

Each exercise (tandem and tiptoe) was performed for 30 s without resting time between exercises and processed for approximately 1 min in total.

#### 2.4.2. Core stability Exercise

As an exercise to stabilize the trunk, CSE consisted of the elbow-toe and hand-heel exercises and was performed as previously described [7]. Each of the two exercises was performed continuously for 30 s and processed for approximately 1 min in total.

To perform elbow-toe exercise, the knees were extended, the elbows bent at 90 degrees in prone position, and the weight was lifted with the toes and the forearm. The hand was made to hold a fist lightly, the neck was in a neutral position, and the trunk, pelvis, and lower extremity were in a straight line. The thoracic region in which the scapular is located is kept flat by protraction during the plank (Figure 4).

To perform the hand-heel exercise, the participant was in a long sitting posture with arms supported and placed heels on a 20 cm-high box. Then, the participant was instructed to raise his hips with elbow extension. The trunk, pelvis, and lower extremity were made to be in a straight line (Figure 5).

Each exercise (elbow-toe and hand-heel) was performed for 30 s without resting time between exercises and processed for approximately 1 min in total.

### 2.5. Outcome Measures

#### 2.5.1. Static Balance Ability

To measure postural sway, a force plate (Zebris PDM, Zebris Medical, Isny, Germany) was used, and the signal was processed using MR3.10 Software (Noraxon, Scottsdale, AZ, USA). During the measurement, the participants took off their shoes and stood straight on the force plate with their arms relaxed comfortably and evenly at their sides. They were instructed to keep their heads as still as possible. Their feet were opened at a 30° angle, and the space between the heels was 9 cm. Participants faced forward and were instructed to stare straight at an X mark (8 cm in width and 6 cm in height) placed at eye level on a wall 1.5 m in front of them [16]. On the researcher’s signal, the measurement started, and force plate data was recorded at 50 Hz.

Recordings were measured for 40 s, and the total measurement of 30 s, which excluded the first and last 5 s, was used for analysis. Additionally, recordings were measured three times each in eye-opened and eye-closed conditions at the baseline and after exercise, and the interval between conditions was set to 10 s. Moreover, as in a previous study, to remove the learning effect, the interval between the measurements at the baseline and after exercise was set to 4 min. To remove the exercise effect, each interval between exercises (CE and CSE) was set to 15 min [7]. The variables measured through the force plate and used in the analysis of static standing balance ability were as follows: 95% confidence ellipse area (mm^2^), center of pressure (COP) path length (mm), COP average velocity (mm/s), length of minor axis (mm), length of major axis (mm), and anterior/posterior foot pressure ratio (A/P ratio). A higher value of the A/P ratio variable means that the center of gravity moved further forward.

#### 2.5.2. Muscle Activation

To measure muscle activation, 8-channel wireless surface EMG was performed using the Noraxon TELEmyo 2400T (NORAXON Inc., Scottsdale, AZ, USA) device. The preparation of measurement was completed by removing hairs around the attachment part before placing the electrode and cleaning with alcohol. Pairs of disposable Ag/AgCl surface electrodes were used, and the distance between electrodes was 2 cm. The electrode was attached around the selected muscle toward the participant’s dominance. For raw EMG signal, the frequency of the band-pass filter was set to 20–350 Hz, and the root mean square (RMS) value was set to 100 ms after rectifying and normalizing with maximal voluntary contraction (MVC). MVC was performed in supine (internal oblique (IO), rectus abdominis (RA), tibialis anterior (TA), medial gastrocnemius (GCM)), prone (erector spinae (ES), biceps femoris (BF)), side-lying (gluteus medius (Gmed)), and sitting (rectus femoris (RF)) positions. MVC values were obtained through trunk flexion and left and right twists for IO and RA, trunk extension for ES, dorsiflexion and plantar flexion for TA and GCM, and hip abduction, knee extension and flexion for Gmed, RF, and BF, respectively [17,18,19,20,21,22]. During the measurement, other segments not related to the target muscle were fixed with a strap and measured by the therapist’s manual resistance. Participants sustained maximum effort contraction for about 3 s, and the measurements were repeated 3 times. In consideration of muscle fatigue, the interval between measurements was set to 30 s.

The selected muscles were the IO (2 cm medial and 2 cm inferior to the anterior superior iliac spine (ASIS)), RA (2 cm lateral to the navel), ES (3 cm lateral to the center of the L3 vertebra), Gmed (midpoint between the lateral border of the iliac crest and greater trochanter), RF (middle of the front thigh, at approximately the midpoint between the knee and ASIS), TA (at the one-third level of the muscle, approximately 15 cm below the knee), GCM (oriented parallel to the muscular fibers), and BF (midpoint between the ischial tuberosity of the femur and lateral epicondyle of the tibia). Electrodes were placed as outlined in a previous study [23,24,25,26,27,28,29]. All evaluations were performed by the same physical therapist with over 5 years of experience, who was blinded to the experiment.

### 2.6. Statistical Analysis

In this study, data are presented as mean and standard deviation and were processed using SPSS 25.0 software (SPSS Inc., Chicago, IL, USA). Normality was assessed using the Shapiro–Wilk test. For within-group comparisons, repeated measures analysis of variance, which is a parametric statistical test, was used for variables with normal distributions. On the other hand, the Friedman test, which is a nonparametric statistical test, was used for variables with non-normal distributions. Post-hoc comparison was performed using the Wilcoxon signed-rank test. The statistical significance level was set to 0.05.

## 3. Results

The characteristics of participants are presented in Table 1.

### 3.1. Static Balance Ability

There was a statistically significant difference in all variables in the eye-opened (Table 2) and eye-closed conditions (*p* < 0.05) (Table 3). As a result of post-hoc comparison, result after CE showed a statistically significant decrease in area, path length, average velocity, minor axis, and major axis and increase in A/P ratio in the eye-opened and eye-closed conditions compared with the baseline values (*p* < 0.05). Result after CSE showed a statistically significant decrease in all variables except the A/P ratio variables in the eye-opened and eye-closed conditions compared with the baseline values (*p* < 0.05). For comparison between exercises, path length and average velocity variables after CE showed a statistically significant decrease in the eye-opened condition compared with those after CSE (*p* < 0.05); in the eye-closed condition, the path length variable after CE showed a statistically significant decrease compared with that after CSE (*p* < 0.05).

### 3.2. Muscle Activation

In the eye-opened condition, a significant difference was noted in the Gmed, RF, TA, and RF/BF ratio (*p* < 0.05) (Table 4). In post-hoc analysis, there was a significant increase in the Gmed and decrease in the RF, TA, and RF/BF ratio compared with baseline values in CE (*p* < 0.05), and the RF, TA, and RF/BF ratio showed a statistically significant decrease compared with baseline values in CSE (*p* < 0.05). There were no significant differences in all variables among exercises (*p* > 0.05).

In the eye-closed condition, a significant difference was noted in the RF and RF/BF ratio (*p* < 0.05) (Table 5). In post-hoc analysis, the RF and RF/BF ratio showed a significant decrease compared with those at the baseline in CE (*p* < 0.05), and the RF showed a significant decrease compared with those at the baseline in CSE (*p* < 0.05). There were no significant differences in all variables among exercises (*p* > 0.05).

## 4. Discussion

This study demonstrated that CE and CSE have a positive impact transiently on healthy male adults’ balance ability and that CE has a more positive impact than CSE. In addition, this study confirmed that both exercises have an impact on muscle activation.

### 4.1. Static Balance Ability

Through CE for 1 min, a significant change in score was observed in the balance ability evaluation in the eye-opened (area, 27.0%; path length, 20.3%; average velocity, 34.9%; minor axis, 13.1%; major axis, 16.1%) and eye-closed (area, 26.9%; path length, 21.5%; average velocity, 17.6%; minor axis, 16.1%; major axis, 11.5%) conditions. In addition, with the eyes open, CE showed a significant decrease in path length and average velocity compared to CSE. Similarly, even with eyes closed, CE showed a significant decrease in path length compared to CSE.

The tandem and tiptoe exercises performed in this study generate instability because the base of support is narrow compared to that in a general standing posture, and weight shifting also generates instability [30]. Halvarsson et al. (2015) mentioned that an unstable situation, such as uneven surface, change in base of support during standing, and reaction to loss of balance, can improve balance factors such as sensory orientation, postural responses, and stability limits [12]. Therefore, it is considered to have improved the static balance ability. In addition, tiptoe and tandem exercises can strengthen postural control through the continuous activation of the calf muscle [11,12]. A study by Spink et al. (2011) reported that plantar flexor strength is associated with postural sway [31], and Judge et al. (2003) mentioned an improvement in standing balance (change score: 21%) through balance training including the tandem stance. Therefore, it is thought that the increase in calf activity influenced static balance ability by the characteristics of CE.

Duclos et al. (2004) reported that isometric muscle contraction for 30 s has an impact on posture [32]. They suggested that this was due to the activation of muscle-related proprioception. In postural control, muscle stiffness is an important factor in responding properly to external perturbation [33]. It is thought that the continuous muscle contraction during exercise used in the study caused temporary muscle stiffness, which improved the static balance ability despite the short period of time. Kaji A et al. (2010) also found improvement in static balance ability through exercise intervention at the same time as this study intervention [7]. This supports the results of this study. This implies that it is suitable as a warm-up exercise for shooting and archery players who require static balance ability. In addition, it is thought that the balance ability will be improved more effectively if this exercise is performed as a warm-up before performing balance training, which is commonly used in clinical settings.

In a comparison between CE and CSE, a significant difference in the path length and average velocity seems to affect exercise composition included in CE. Kaji et al. (2010) mentioned that although CSE had a positive effect on static balance, there was no significant difference in postural sway in the anteroposterior direction [7]. In the exercises used in this study, the weight shifting of the tandem exercise and tiptoe exercise positively influenced the control in the antero-posterior direction [10], and the ability to control static balance in the frontal plane during the tandem exercise was also required. As such, in this study, the more significant improvement of static balance ability in CE is thought to be due to a positive effect on the postural sway of the AP direction, which could not be strengthened through CSE. In addition, it seems that there was a difference in the characteristics of the exercise task.

### 4.2. Muscle Activation

In the eye-opened and eye-closed conditions for CE, there was a significant decrease in the RF and RF/BF ratio. In order to maintain static balance, the center of gravity (COG) must be located within the base of support (BOS) [34,35,36]. The movement of COG causes the activation of the muscle opposite to the direction of movement in static standing. Based on a significant increase in the A/P ratio, it is speculated that the decrease in RF activation was caused by the anterior movement of COG after CE (Table 2). In addition, the activity of BF did not show a significant increase, but caused by a factor of decreasing RF activation, the RF/BF ratio seems to have decreased.

In the eye-opened condition, we identified an increase in Gmed activation after CE. In the case of CSE, base of support is wide, and there is upper limb intervention due to exercise in a supine or prone position. Contrarily, CE is an exercise requiring only the activity of the lower extremity muscles and continuous activity of the Gmed. Therefore, after CE, it seems that the increase in Gmed activation is attributable to the exercise characteristic, and Gmed activation has an impact on balance ability’s improvement.

This study has limitations as follows. First, all participants were men; thus, the results cannot be generalized to women. Second, we cannot suggest a specific interpretation of results because there are no studies that provide a reference value for the hamstring/quadriceps ratio. Therefore, a study that can suggest the normal value is needed. Third, the exercise’s long-term effect is uncertain. Future studies should assess the effects of long-term exercise on static standing balance ability, analyze dynamic movement, include athletes who require static standing balance ability as participants, and investigate a more effective exercise program.

## 5. Conclusions

This study showed that the CE can immediately improve the static balance ability and bring about changes in muscle activity in healthy male adults. It also showed a greater improvement in static balance ability than the CSE, which is commonly used in clinical settings. This result showed the possibility of applying CE to activities of daily living or to those of sports athletes without space constraints. In particular, it is recommended to apply CE as training for healthy persons who enjoy surfing, shooting, and archery as leisure activities or as pre-match exercise for sports players who require static balance skills.

## Figures and Tables

**Figure 1 healthcare-08-00375-f001:**
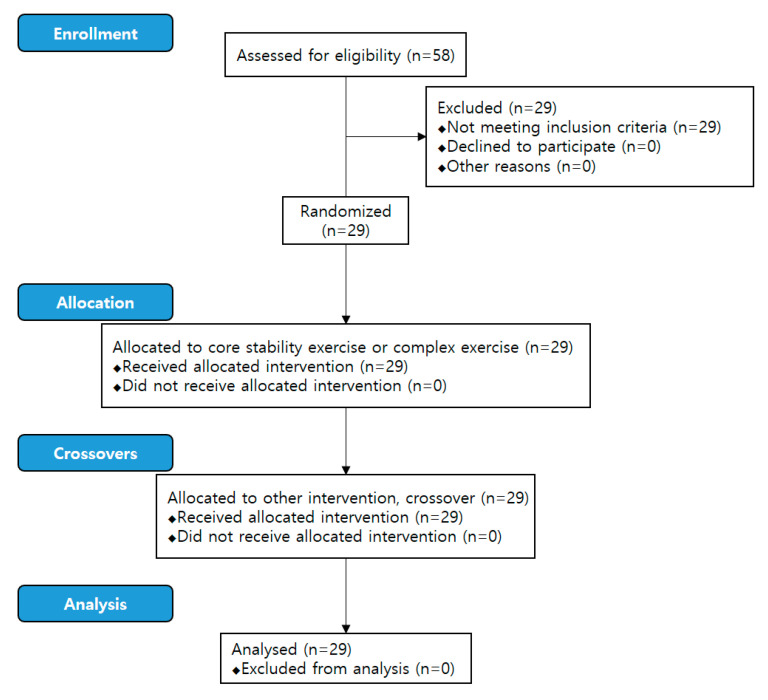
CONSORT flow chart.

**Figure 2 healthcare-08-00375-f002:**
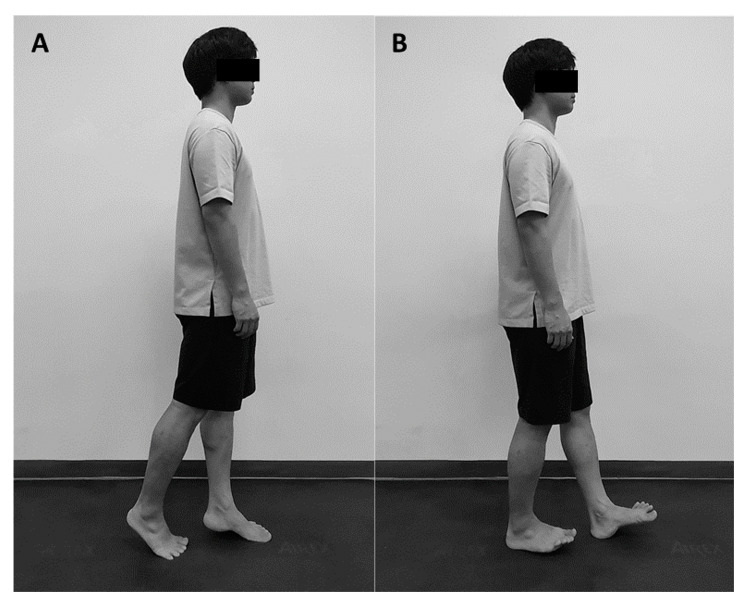
Tandem exercise. (**A**) Forward movement; (**B**) Backward movement.

**Figure 3 healthcare-08-00375-f003:**
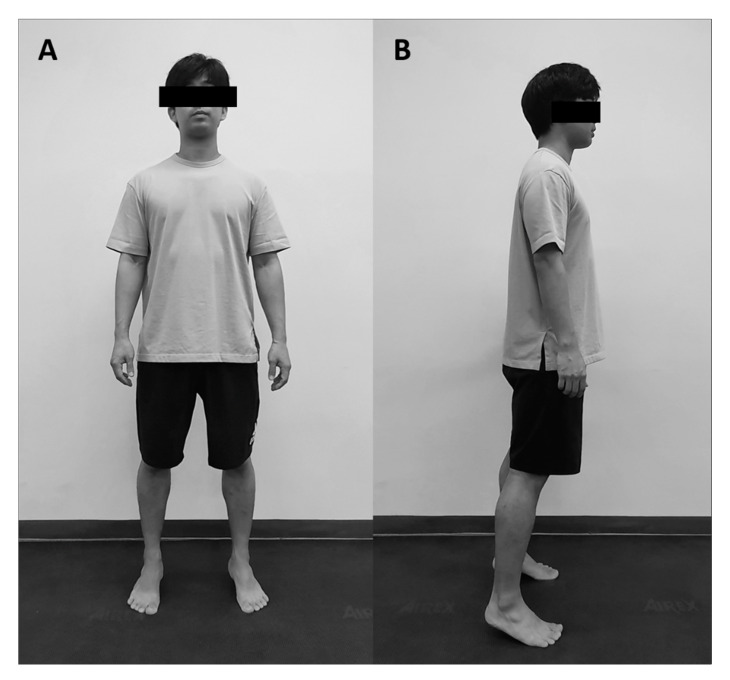
Tiptoe exercise. (**A**) Front view; (**B**) Lateral view.

**Figure 4 healthcare-08-00375-f004:**
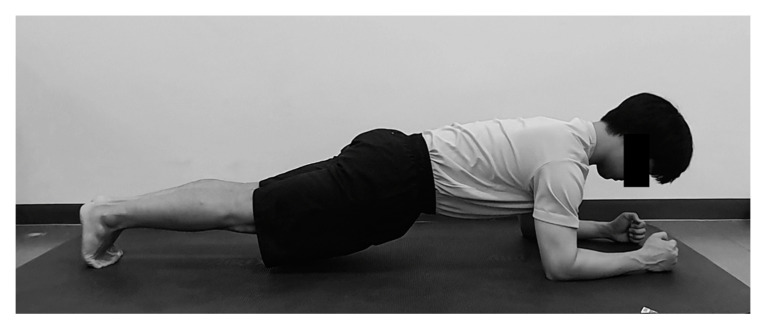
Elbow-toe exercise.

**Figure 5 healthcare-08-00375-f005:**
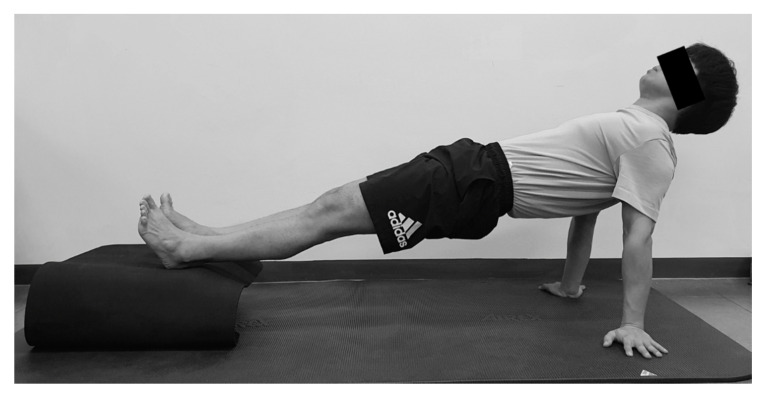
Hand-heel exercise.

**Table 1 healthcare-08-00375-t001:** Characteristics of participants.

Variables	Mean ± SD
Age (y)	25.17 ± 3.24
Height (cm)	173.58 ± 5.25
Weight (kg)	69.53 ± 11.23
BMI (kg/m^2^)	23.02 ± 3.08

Values are expressed as mean ± SD. Abbreviation: BMI, body mass index.

**Table 2 healthcare-08-00375-t002:** Comparison of static balance ability in the eye-opened condition.

Variables	B	CSE	CE	*p* Value	Post Hoc Analysis		
					B-CSE	B-CE	CSE-CE
Area (mm^2^)	127.08 ± 50.75	100.29 ± 40.43	92.82 ± 43.38	<0.001 *	<0.001 ^#^	<0.001 ^#^	0.294
Path length (mm)	169.35 ± 46.15	146.59 ± 33.55	134.90 ± 40.81	<0.001 *	<0.001 ^#^	<0.001 ^#^	0.013 ^#^
Average velocity (mm/s)	5.65 ± 1.54	4.96 ± 1.15	4.68 ± 1.29	<0.001 *	0.001 ^#^	<0.001 ^#^	0.041 ^#^
Minor axis (mm)	8.42 ± 2.55	7.42 ± 2.08	7.32 ± 2.86	0.001 *	0.004 ^#^	0.002 ^#^	0.489
Major axis (mm)	19.67 ± 4.51	16.89 ± 3.48	16.51 ± 4.50	0.001 *	0.016 ^#^	0.005 ^#^	1
A/P ratio (%)	44.64 ± 7.50	45.86 ± 7.04	47.06 ± 7.91	0.036 *	0.187	0.002 ^#^	0.186

Values are expressed as mean ± SD. Abbreviation: B, baseline; CSE, core stability exercise; CE, complex exercise; A/P ratio, anterior/posterior foot pressure ratio. * Significant differences (*p* < 0.05) between three different conditions; # significant differences (*p* < 0.05) between two conditions.

**Table 3 healthcare-08-00375-t003:** Comparison of static balance ability in the eye-closed condition.

Variables	B	CSE	CE	*p* Value	Post Hoc Analysis		
					B-CSE	B-CE	CSE-CE
Area (mm^2^)	165.34 ± 89.11	121.07 ± 62.54	120.81 ± 59.56	<0.001 *	<0.001 ^#^	<0.001 ^#^	0.837
Path length (mm)	221.81 ± 69.54	186.48 ± 52.89	174.18 ± 56.26	<0.001 *	<0.001 ^#^	<0.001 ^#^	0.025 ^#^
Average velocity (mm/s)	7.38 ± 2.34	6.32 ± 1.85	6.08 ± 1.82	<0.001 *	<0.001 ^#^	<0.001 ^#^	0.127
Minor axis (mm)	9.47 ± 3.41	8.23 ± 3.29	7.95 ± 2.55	0.003 *	0.004 ^#^	0.002 ^#^	0.489
Major axis (mm)	21.32 ± 5.05	19.11 ± 4.82	18.86 ± 4.97	0.006 *	0.041 ^#^	0.015 ^#^	1
A/P ratio (%)	46.48 ± 6.66	47.22 ± 6.81	48.68 ± 7.18	0.021 *	0.254	0.003 ^#^	0.121

Values are expressed as mean ± SD. Abbreviation: B, baseline; CSE, core stability exercise; CE, complex exercise; A/P ratio, anterior/posterior foot pressure ratio. * Significant differences (*p* < 0.05) between three different conditions; # significant differences (*p* < 0.05) between two conditions.

**Table 4 healthcare-08-00375-t004:** Comparison of muscle activation in the eye-opened condition.

Variables	B	CSE	CE	*p* Value	Post Hoc Analysis		
					B-CSE	B-CE	CSE-CE
Internal oblique	19.72 ± 13.34	19.43 ± 13.09	18.23 ± 13.84	0.639	0.596	0.078	0.37
Rectus abdominis	10.10 ± 1.74	9.36 ± 1.40	9.66 ± 1.40	0.316	0.187	0.429	0.399
Erector spinae	8.35 ± 4.21	8.81 ± 5.78	9.40 ± 6.19	0.381	0.863	0.23	0.596
Gluteus medius	6.90 ± 8.46	7.63 ± 8.41	8.66 ± 10.29	0.004 *	0.111	0.003 ^#^	0.055
Rectus femoris	15.98 ± 13.00	13.45 ± 11.37	13.75 ± 12.10	0.001 *	0.003 ^#^	<0.001 ^#^	0.604
Tibialis anterior	16.61 ± 16.12	14.30 ± 13.69	14.51 ± 14.34	0.012 *	0.033 ^#^	0.005 ^#^	0.983

Values are expressed as mean ± SD. Abbreviation: B, baseline; CSE, core stability exercise; CE, complex exercise; RF, rectus femoris; BF, biceps femoris; TA, tibialis anterior; GCM, gastrocnemius. * Significant differences (*p* < 0.05) between three different conditions; # significant differences (*p* < 0.05) between two conditions.

**Table 5 healthcare-08-00375-t005:** Comparison of muscle activation in the eye-closed condition.

Variables	B	CSE	CE	*p* Value	Post Hoc Analysis		
					B-CSE	B-CE	CSE-CE
Internal oblique	20.09 ± 15.55	19.42 ± 13.55	18.51 ± 13.68	0.343	0.336	0.098	0.611
Rectus abdominis	10.09 ± 9.01	9.90 ± 7.64	9.95 ± 7.55	0.073	0.247	0.122	0.804
Erector spinae	8.20 ± 4.30	9.35 ± 7.04	9.92 ± 7.59	0.966	0.787	0.336	0.82
Gluteus medius	6.35 ± 6.19	6.91 ± 6.14	7.97 ± 9.39	0.576	0.347	0.265	0.627
Rectus femoris	15.87 ± 13.21	14.18 ± 12.36	13.83 ± 12.68	<0.001 *	0.002 ^#^	<0.001 ^#^	0.91
Tibialis anterior	15.93 ± 15.51	14.49 ± 13.44	14.75 ± 14.77	0.185	0.163	0.142	0.846

Values are expressed as mean ± SD. Abbreviation: B, baseline; CSE, core stability exercise; CE, complex exercise; RF, rectus femoris; BF, biceps femoris; TA, tibialis anterior; GCM, gastrocnemius. ^*^ Significant differences (*p* < 0.05) between three different conditions; ^#^ significant differences (*p* < 0.05) between two conditions.

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
