# Peer review of "A Comparison of the Transient Effect of Complex and Core Stability Exercises on Static Balance Ability and Muscle Activation during Static Standing in Healthy Male Adults"

_healthcare, 2020, doi:10.3390/healthcare8040375_

Round 1
Reviewer 1 Report
Overall the paper is very good and comprehensive but it is a bit challenging to read. I have a few suggestions: (1) add addreviations to the phases in the keywords section, (2) if possible include picture of the various exercises to help visualize what you are talking about. Also, consider expanding the conclusions to summarize the top 5 findings and more detail on the benefits.
Author Response
Comments from Reviewer 1:
- Overall the paper is very good and comprehensive but it is a bit challenging to read. I have a few suggestions: (1) add abbreviations to the phases in the keywords section
Response:
- Thank you for your interesting suggestion. As per your recommendation, we additionally insert abbreviations for each key words as follow:
- In line 32-33: “Keywords: Postural sway; Complex exercise (CE); Balance; Muscle activation; Core stability exercise (CSE)”
- if possible include picture of the various exercises to help visualize what you are talking about. Also, consider expanding the conclusions to summarize the top 5 findings and more detail on the benefits.
Response:
- You have made a good suggestion, and we have added a picture of the exercise method used in this study accordingly.
- In line 118: “Figure 2. Tandem exercise. A, Forward movement; B, Backward movement.”
- In line 125: “Figure 3. Tiptoe exercise. A, Front view; B, Lateral view.”
- In line 138: “Figure 4. Elbow-toe exercise.”
- In line 143: “Figure 5. Hand-heel exercise.”
- All authors appreciated your valuable comments and agree for this issue. We have summarized our findings, and in particular, describe the benefits our results can be used in clinics as follows:
- In line 311-316: “This study showed that the CE can immediately improve the static balance ability and bring about changes in muscle activity in healthy male adults. It also showed a greater improvement in static balance ability than the CSE, which is commonly used in clinical setting. This result showed the possibility of applying CE to activities of daily living or sports athletes without space constraints. In particular, it is recommended to apply CE as a training of healthy person who enjoy surfing, shooting, and archery as leisure or pre-match exercise to sports players, which require static balance skills.”

Reviewer 2 Report
This is a study of pre-exercise stability exercises and their effect on muscle activation and balance stability. Interesting results. I believe the methods have been done well, however the narrative description is difficult to understand . perhaps this is English translation related? Please see comments below.
Abstract:
- The abstract mentions significant differences, but does not provide indication of direction of difference. Perhaps indicate direction (i.e “significant reduction in path length & vertical velocity..)
- Line 29 I would remove the sentence “In particular it showed bigger significant difference in CE”
“it” is not clear, and direction of change is not stated.
- Line 30 Muscle activation was significantly “affected”. Again, please provide a direction of change. Reduced activation? Increased?
Material and Methods
Line 64- you mention 35 subjects: line 76 : 29 subjects , line 81: 58 subjects. This is very confusing to the reader. Perhaps only mention subject selection in one sentence at the beginning of this section. Then ref the reader to Figure 1.
2.4.1
This section is quite confusing. It may be helpful to have some sort of image showing the movement. I was not able to create a visual of what the subject was doing as the treatment.
- Line 97 “This maneuver”. Unlcear which one you are referring to. Tiptoe? Tandem?
- Line 100- what is “lateral expansion”. No body position or section is indicated
Suggest 2.4.1 First label and explain the bracing maneuver., Next label and explain Tiptoe, and tandem maneuver. Describe the nature of the weight shifting (joints that move, upper body flexion or extension?) The reader should be able to replicate the treatment based on your description.
2.4.2
CE- While I understand the ref. 10 has a longer description of the exercise, there should be enough of a description in this manuscript that one can visualize the stability exercise.
2.5.2 Muscle Activation
- Identify that you used surface EMG. Interelectrode distance? Unipolar or bipolar?
Please provide information on your normalizing procedure. Was this against a brace by the therapist? Different types of movements for each muscle…
Results
In your results you describe the changes by stating “ there were statistical differences”. This does not provide the reader any information on the direction of the changes. Was one condition higher than baseline? Lower? Provide direction of the change in your narrative. The reader can look at the tables, but you should be indicating the direction of change in the narrative.
Discussion
4.1 The results are showing comparisons across the treatments, but separately for eyes open (table 2) and eyes closed (table 3).
However your narrative line 229 on seem to compare eyes open to closed. I would revise the discussion to match the analysis. Can it perhaps provide information and previous literature on how such a brief exposer to either CSE or CE cause such a significant improvement in balance ability? This would be most interesting to the reader. Think in terms of practical application for the therapist (pre-exercise warm up prior to balance activity)
4.2 Muscle activation
Please provide more information about why there was a significant decrease in the RF activation. Line 244 indicates that “it seems that this is because of the forward movement of the body’s center. Are these results you have in a table somewhere?
Line 253 You imply that the changed RF/BF ratio increased stability. How can you tell if it is responsible for the change or is merely an outcome of improved balance (since the RF is activated less).
Author Response
Comments from Reviewer 2:
Abstract
- The abstract mentions significant differences, but does not provide indication of direction of difference. Perhaps indicate direction (i.e “significant reduction in path length & vertical velocity.)
Response:
- As the Reviewer recommended, the details about direction of difference were additionally described.
- In line 23-30: “In comparisons among exercises, the path length and average velocity variables showed a significant decrease in the eye-opened condition, and the path length variable showed a significant decrease in the eye-closed condition (P<0.05). In muscle activation, CE showed a significant increase in the gluteus medius (Gmed) and decrease in the rectus femoris (RF), tibialis anterior (TA), and RF/biceps femoris (BF) ratio in the eye-opened condition compared to baseline and a significant decrease in RF and RF/BF ratio in the eye-closed condition (P<0.05). Both CE and CSE improved the static balance ability. Furthermore, muscle activation was significantly increases in Gmed and decreases in RF, TA, RF/BF ratio.”
- Line 29 I would remove the sentence “In particular it showed bigger significant difference in CE” “it” is not clear, and direction of change is not stated.
Response:
- All authors appreciate your point. We removed this sentence as per your suggestion.
- Line 30 Muscle activation was significantly “affected”. Again, please provide a direction of change. Reduced activation? Increased?
Response:
- All authors appreciate the Reviewer’s points. As you pointed-out, we modified this issue as follow:
- In line 29-30: “Furthermore, muscle activation was significantly increases in Gmed and decreases in RF, TA, RF/BF ratio.”
Material and Methods
- Line 64- you mention 35 subjects: line 76: 29 subjects , line 81: 58 subjects. This is very confusing to the reader. Perhaps only mention subject selection in one sentence at the beginning of this section. Then ref the reader to Figure 1.
Response:
- We mentioned the total number of subjects who applied for the study and the number of people who ultimately performed it. I thought that this could confuse the reader, so we have revised the content as follows:
- In line 65-67: “This study was conducted for 3 months at Gachon University. 58 university students responded to a campus announcement and 29 met the inclusion criteria and participated in the experiment (Figure 1).”
- And we remove the sentences for the experimental procedures.
- In line 81-83: “Fifty-eight individuals volunteered to participate in this study, and after excluding 29 who did not meet the inclusion criteria, a total of 29 participants were included. The CONsolidated Standards of Reporting Trials (CONSORT) flow chart was shown in the Figure 1.”
- 4.1 Complex exercise: This section is quite confusing. It may be helpful to have some sort of image showing the movement. I was not able to create a visual of what the subject was doing as the treatment.
Response:
- Authors would like to thanks the Reviewer for this recommendation. In accordance with your suggestion, we inserted the Figure showing the exercise movements performed in the study.
- In line 118: “Figure 2. Tandem exercise. A, Forward movement; B, Backward movement.”
- In line 125: “Figure 3. Tiptoe exercise. A, Front view; B, Lateral view.”
- Line 97 “This maneuver”. Unlcear which one you are referring to. Tiptoe? Tandem?
Response:
- Thanks for your advice. This maneuver means the bracing maneuver, and we modified it to be bracing maneuver:
- In line 99-101: “Bracing maneuver. It was performed by instructing them to “focus on the abdominal muscles and hold their breath after exhalation” and then maintain the abdominal isometric contraction.”
- Line 100- what is “lateral expansion”. No body position or section is
Response:
- Thanks very much for your observation. We agree with Reviewer’s comment and additionally described the "lateral expansion" for the reader's understanding as follow:
- In line 101-104: “For the proper performance of the bracing maneuver, a researcher confirmed core muscle contraction and trunk lateral expansion through palpation between the 12th rib and iliac crest in the axillary line before starting the exercise.”
- Suggest 2.4.1 First label and explain the bracing maneuver., Next label and explain Tiptoe, and tandem maneuver. Describe the nature of the weight shifting (joints that move, upper body flexion or extension?) The reader should be able to replicate the treatment based on your description.
Response:
- Thank you for your detailed point. To make it easier for readers to understand and follow the exercise interventions used in the study, we have categorized and explained these as follows:
- In line 99-105: “Bracing maneuver. It was performed by instructing them to “focus on the abdominal muscles and hold their breath after exhalation” and then maintain the abdominal isometric contraction [15]. For the proper performance of the bracing maneuver, a researcher confirmed core muscle contraction and trunk lateral expansion through palpation between the 12th rib and iliac crest in the axillary line before starting the exercise. To maintain bracing maneuver during tandem exercise and tiptoe exercise, the therapist confirmed the EMG activity and provided verbal cue to the participant.”
- In line 106-116: “Tandem exercise. The participant made a tandem stance posture with the gaze at the front, arms at the pelvis side, and a minimum distance of 5 cm between the front and rear feet. Weight shifting was performed in an anteroposterior direction like a pendulum movement while taking the tandem stance. Weight shifting was performed at a constant rhythm using a smartphone metronome application (Tempo Lite, Frozen Ape Pte. Ltd., Singapore) by adjusting to participants’ preferred speed (1-2 Hz). During weight shifting, the weight shifting range was moved until only the first and second metatarsal heads of both feet touched the ground when moving forward, and only the calcaneus of both feet touched when moving backward. The trunk maintained the bracing maneuver without flexion or extension movement (Figure 2). This exercise performed for 30 seconds, participant continued to weight shifting in an anteroposterior direction.”
- In line 119-123: “Tiptoe exercise. The participant stood upright with feet shoulder-width apart, staring straight ahead and arms were positioned at the sides. Then, the participant was only instructed to raise heel. During exercise, the therapist instructed the bracing maneuver to be maintained (Figure 3). This exercise performed for 30 seconds, participant kept heel out of contact with the ground.”
- And additionally we described the sentence about weight shifting as follows:
- In line 108-115: “Weight shifting was performed in an anteroposterior direction like a pendulum movement while taking the tandem stance. Weight shifting was performed at a constant rhythm using a smartphone metronome application (Tempo Lite, Frozen Ape Pte. Ltd., Singapore) by adjusting to participants’ preferred speed (1-2 Hz). During weight shifting, the weight shifting range was moved until only the first and second metatarsal heads of both feet touched the ground when moving forward, and only the calcaneus of both feet touched when moving backward. The trunk maintained the bracing maneuver without flexion or extension movement (Figure 2).”
- 4.2 core stability exercise: CE- While I understand the ref. 10 has a longer description of the exercise, there should be enough of a description in this manuscript that one can visualize the stability exercise.
Response:
- We acknowledge the Reviewer’s recommendation and additionally described a more detailed description of the interventional exercises performed in this study as follows:
- In line 132-136: “To perform elbow-toe exercise, the knees were extended and the elbows were bent at 90 degrees in prone position, and the weight was lifted with toe and the forearm. The hand was made to hold a fist lightly, the neck was in a neutral position, and the trunk, pelvis, and lower extremity were in a straight line. The thoracic region in which the scapular is located is kept flat by protraction during the plank (Figure 4).”
- In line 139-141: “To perform the hand-heel exercise, participant was in a long sitting posture with arms supported and placed heels on the 20cm-high box. Then, participant was instructed to raise hip with elbow extension. The trunk, pelvis, and lower extremity were made to be in a straight line (Figure 5).”
- In line 144-145: “Each exercise (elbow-toe and hand-heel) was performed for 30 seconds without resting time between exercises and processed for approximately 1 min in total.”
- And insert the Figure:
- In line 138: “Figure 4. Elbow-toe exercise.”
- In line 143: “Figure 5. Hand-heel exercise.”
- 5.2 Muscle Activation - Identify that you used surface EMG. Interelectrode distance? Unipolar or bipolar?
Response:
- As you recommended, we additionally described this as follow:
- In line 168-169: “To measure muscle activation, 8-channel wireless surface EMG was performed using the Noraxon TELEmyo 2400T (NORAXON Inc., Scottsdale, AZ, USA) device.”
- In line 171-172: “Pairs of disposable Ag/AgCl surface electrodes were used, and the distance between electrodes was 2 cm.”
- Please provide information on your normalizing procedure. Was this against a brace by the therapist? Different types of movements for each muscle…
Response:
- As the reviewer’s pointed out, we additionally described this issue as follow:
- In line 175-183: “MVC was performed in supine (internal oblique (IO), rectus abdominis (RA), tibialis anterior (TA), medial gastrocnemius (GCM)), prone (erector spinae (ES), biceps femoris (BF)), side-lying (gluteus medius (Gmed)), and sitting (rectus femoris (RF)) positions. MVC values were obtained through trunk flexion and left and right twist for IO and RA, trunk extension for ES, dorsiflexion and plantar flexion for TA and GCM, and hip abduction, knee extension and flexion for Gmed, RF, and BF, respectively [17-22]. During the measurement, other segments not related to the target muscle were fixed with a strap and measured by the therapist's manual resistance. Participants sustained maximum effort contraction for about 3 seconds, and the measurements were repeated 3 times. In consideration of muscle fatigue, the interval between measurements was set to 30 seconds.”
Results
- In your results you describe the changes by stating “there were statistical differences”. This does not provide the reader any information on the direction of the changes. Was one condition higher than baseline? Lower? Provide direction of the change in your narrative. The reader can look at the tables, but you should be indicating the direction of change in the narrative.
Response:
- In accordance with the Reviewer's recommendation, we provided the direction of the change in the Results section as follows:
- In line 205-213: “As a result of post hoc comparison, result after CE showed a statistically significant decrease in area, path length, average velocity, minor axis, major axis and increase in A/P ratio in the eye-opened and eye-closed conditions compared with the baseline values (P<0.05). Result after CSE showed a statistically significant decrease in all variables except the A/P ratio variables in the eye-opened and eye-closed conditions compared with the baseline values (P<0.05). For comparison between exercises, path length and average velocity variables after CE showed a statistically significant decrease in the eye-opened condition compared with those after CSE (P<0.05); in the eye-closed condition, path length variable after CE showed a statistically significant decrease compared with that after CSE (P<0.05).”
- In line 228-231: “In post hoc analysis, there was a significant increase in Gmed and decrease in RF, TA, and RF/BF ratio compared with baseline values in CE (P<0.05), and RF, TA, and RF/BF ratio showed a statistically significant decrease compared with baseline values in CSE (P<0.05). There were no significant differences in all variables among exercises (P>0.05).”
- In line 239-241: “In post hoc analysis, RF and RF/BF ratio showed a significant decrease compared with those at baseline in CE (P<0.05), and RF showed a significant decrease compared with those at baseline in CSE (P<0.05). There were no significant differences in all variables among exercises (P>0.05).”
Discussion
- 1 The results are showing comparisons across the treatments, but separately for eyes open (table 2) and eyes closed (table 3). However your narrative line 229 on seem to compare eyes open to closed. I would revise the discussion to match the analysis.
Response:
- We acknowledge the Reviewer’s recommendation. In order to interpret the results of this study in various ways, we compared the results of eye opened condition and eye closed condition in 1st However, we did not describe this in the Methods and Results sections, and we judged this to be inappropriate to describe in the Discussion section. Thus, we decided that this paragraph was not necessary and deleted it. Thanks again for the good point.
- And did not present your point in the Discussion section. Once again, thank you for your good point. We have additionally described the following:
- In line 256-258: “In addition, with the eyes open, CE showed a significant decrease in path length and average velocity compared to CSE. Similarly, even with eyes closed, CE showed a significant decrease in path length compared to CSE.”
- Can it perhaps provide information and previous literature on how such a brief exposer to either CSE or CE cause such a significant improvement in balance ability? This would be most interesting to the reader. Think in terms of practical application for the therapist (pre-exercise warm up prior to balance activity)
Response:
- We acknowledge that the Reviewer's suggestion is a very necessary matter for this section. As you may well know, many studies to date have reported that exercise interventions improved balance and motor function in different types of subjects (those with nervous or musculoskeletal problems). However, these studies performed exercise intervention over a long period of time or applied multiple times to subjects. While these results are very valuable, there are some limitations.
- First, for exercise intervention to be effective on balance or motor function, it must be repeated several times. Second, repeated participation of the participant is essential to achieve this result. We experienced that a single or a short number of exercise interventions also showed this effect in the process of treating patients, and this study was designed to prove this issue.
- And to our knowledge, there are currently no studies that have elucidated the effects of a single exercise intervention on static balance and muscle activation. This study was designed to clarify this issue, and is also the first attempt to report these results. For this reason, it is difficult to present references requested by the reviewer. Reviewer may be disappointed in this area, but we hope you understand this situation. And we additionally described as follow:
- In line 270-279: “Duclos et al. (2004) reported that isometric muscle contraction for 30 seconds has an impact on posture [32]. They suggested that this was due to the activation of muscle-related proprioception. In postural control, muscle stiffness is an important factor in responding properly to external perturbation [33]. It is thought that the continuous muscle contraction during exercise used in the study caused temporary muscle stiffness, which improved the static balance ability despite a short period of time. Kaji A et al. (2010) also found improvement in static balance ability through exercise intervention at the same time as this study intervention [7]. This supports the results of this study. This implies that it is suitable as a warm-up exercise for shooting and archery players who require static balance ability. In addition, it is thought that the balance ability will be improved more effectively if this exercise is performed warm up before performing balance training, which is commonly used in clinical setting.”
- 2 Muscle activation: Please provide more information about why there was a significant decrease in the RF activation.
Response:
- The reviewer made a good suggestion to us, and this point is very helpful in improving the reader's understanding and quality of the paper. We have additionally described this content as follows:
- In line 290-295: “In the eye-opened and eye-closed conditions, for CE, there was a significant decrease in RF and RF/BF ratio. In order to maintain static balance, the center of gravity (COG) must be located within the base of support (BOS). The movement of COG causes the activation of the muscle opposite to the direction of movement in static standing. Based on a significant increase in the A/P ratio, it is speculated that the decrease in RF activation was caused by the anterior movement of COG after CE (Table 2).”
- Line 244 indicates that “it seems that this is because of the forward movement of the body’s center. Are these results you have in a table somewhere?
Response:
- We judged that 'the forward movement of the body's center' was induced in the subject through the change in A/P ratio variable. For the reader's easy understanding, we described the following additionally, and also presented the corresponding the Table:
- In line 165-165: “The higher value of the A/P ratio variable means that the center of gravity moved more forward.”
- In line 293-295: “Based on a significant increase in the A/P ratio, it is speculated that the decrease in RF activation was caused by the anterior movement of COG after CE (Table 2).”
- Line 253 You imply that the changed RF/BF ratio increased stability. How can you tell if it is responsible for the change or is merely an outcome of improved balance (since the RF is activated less).
Response:
- We acknowledge the Reviewer’s concerns and thought that our expression could confuse the Reviewer and the reader. We have revised the content as follows:
- In line 290-296: “In the eye-opened and eye-closed conditions, for CE, there was a significant decrease in RF and RF/BF ratio. In order to maintain static balance, the center of gravity (COG) must be located within the base of support (BOS) [28-30]. The movement of COG causes the activation of the muscle opposite to the direction of movement in static standing. Based on a significant increase in the A/P ratio, it is speculated that the decrease in RF activation was caused by the anterior movement of COG after CE (Table 2). In addition, the activity of BF did not show a significant increase, but cause by a factor of decreasing RF activation, the RF/BF ratio seems to be decreased.”

Reviewer 3 Report
Dear Authors ,
I thank the researchers for submitting the paper, but there are many comments:
Firstly, the researchers did not mention anything about the type of balance if it is static or dynamic balance!
Only at the end of the study I got it as a ststic balance, while the sports mentioned in the introduction are sports that need to measure the moving balance and improve it. The researchers did not mention any information about the how to filter the EMG data, and whether there is a synchronization with the measurement on the force platform or not!
Was the measurement done separately?
The researchers should explain the training program used in a schedule that includes repetitions, times, intensity, rest periods and intensity of training!
As well as to explain the reasons for choosing exercises!
For statistics and the results: Why the results were not displayed changes within the groups? Considering that what was displayed in the tables are between Groups! As well as why the magnitude of the effect size and the values of the cohen's d and the extent of the strength of the statistical test using Omega Square were not displayed! I need to see the new version before making a decision.
Author Response
Comments from Reviewer 3:
- Firstly, the researchers did not mention anything about the type of balance if it is static or dynamic balance! Only at the end of the study I got it as a static balance, while the sports mentioned in the introduction are sports that need to measure the moving balance and improve it.
Response:
- Thank you for your detailed points. We substituted the main Title as well as table titles as follows:
- In line 2-5: “A Comparison of the Transient Effect of Complex and Core Stability Exercises on Static Balance Ability and Muscle Activation During Static Standing in Healthy Male Adults”
- In line 148: “2.5.1. Static balance ability”
- In line 203: “3.1. Static balance ability”
- In Table 2: “Comparison of static balance ability in eye-opened condition.”
- In Table 3: “Comparison of static balance ability in eye-closed condition.”
- In line 252: “4.1. Static balance ability”
- In addition, we additionally describe the content for the static balance in the Introduction section as follows:
- In line 38-41: “Furthermore, in modern days, because of increased participation in leisure sports that requires static standing balance such as surfing, snowboarding, archery, and shooting, and as the delicate work has increased, balance training is also emphasized to improve sports playing ability and work efficiency [2,3].”
- The researchers did not mention any information about the how to filter the EMG data, and whether there is a synchronization with the measurement on the force platform or not! Was the measurement done separately?
Response:
- I appreciate reviewer’s advice. We agree with reviewer’s comment and additionally described the contents for this as follow:
- In line 85-86: “The surface EMG and force plate were synchronized.”
- In line 173-175: “For raw EMG signal, the frequency of the band-pass filter was set to 20–350 Hz, and the root mean square (RMS) value was set to 100 ms after rectifying and normalizing with maximal voluntary contraction (MVC).”
- In line 175-183: “MVC was performed in supine (internal oblique (IO), rectus abdominis (RA), tibialis anterior (TA), medial gastrocnemius (GCM)), prone (erector spinae (ES), biceps femoris (BF)), side-lying (gluteus medius (Gmed)), and sitting (rectus femoris (RF)) positions. MVC values were obtained through trunk flexion and left and right twist for IO and RA, trunk extension for ES, dorsiflexion and plantar flexion for TA and GCM, and hip abduction, knee extension and flexion for Gmed, RF, and BF, respectively. During the measurement, other segments not related to the target muscle were fixed with a strap and measured by the therapist's manual resistance. Participants sustained maximum effort contraction for about 3 seconds, and the measurements were repeated 3 times. In consideration of muscle fatigue, the interval between measurements was set to 30 seconds.”
- The researchers should explain the training program used in a schedule that includes repetitions, times, intensity, rest periods and intensity of training! As well as to explain the reasons for choosing exercises!
Response:
- We added the followings to the Methods section according to the Reviewer's points:
- In line 109-111: “Weight shifting was performed at a constant rhythm using a smartphone metronome application (Tempo Lite, Frozen Ape Pte. Ltd., Singapore) by adjusting to participants’ preferred speed (1-2 Hz).”
- In line 115-116: “This exercise performed for 30 seconds, participant continued to weight shifting in an anteroposterior direction”
- In line 122-123: “This exercise performed for 30 seconds, participant kept heel out of contact with the ground.”
- In line 126-127: “Each exercise (tandem and tiptoe) was performed for 30 seconds without resting time between exercises and processed for approximately 1 min in total.”
- We selected the exercise used in the study based on the previous literatures. There are two important considerations when choosing an exercise. First, exercise must be able to improve the subject's static balance ability. Second, the subject should not need a lot of space to perform the exercise. This is because the subject could be able to perform the exercise even in situations where there is a space constraint. We described these in the Materials and Methods section as follows:
- In line 96-97: “The composed exercises were selected as exercises that can improve balance and are easy to perform without space constraints.”
- For statistics and the results: Why the results were not displayed changes within the groups? Considering that what was displayed in the tables are between Groups! As well as why the magnitude of the effect size and the values of the cohen's d and the extent of the strength of the statistical test using Omega Square were not displayed! I need to see the new version before making a decision.
Response:
- Thank you for the good point. We conducted the experiment as a single group, and all subjects who participated in this study performed both CSE and CE. All authors checked the manuscript after receiving the point of the review, and it was considered that the contents of the result were not well described in 1st Thus, we modified this as follows:
- In line 205-213: “As a result of post hoc comparison, result after CE showed a statistically significant decrease in area, path length, average velocity, minor axis, major axis and increase in A/P ratio in the eye-opened and eye-closed conditions compared with the baseline values (P<0.05). Result after CSE showed a statistically significant decrease in all variables except the A/P ratio variables in the eye-opened and eye-closed conditions compared with the baseline values (P<0.05). For comparison between exercises, path length and average velocity variables after CE showed a statistically significant decrease in the eye-opened condition compared with those after CSE (P<0.05); in the eye-closed condition, path length variable after CE showed a statistically significant decrease compared with that after CSE (P<0.05).”
- Additionally, we revised the ‘Figure 1. CONSORT flow chart’ to make it easier for readers as well as the Reviewers to understand our experimental process.
And we tried to add effect size data in the ‘2.6. statistical analysis’ section and the Table, but due to the point out of other reviewers, the discussion contents of that part were completely deleted. So we did not add it in the '2.6. statistical analysis' section. We apologize for not being able to correct your point satisfactorily, and we ask you to understand this situation.
